# H2AJ Is a Direct Androgen Receptor Target Gene That Regulates Androgen-Induced Cellular Senescence and Inhibits Mesenchymal Markers in Prostate Cancer Cells

**DOI:** 10.3390/cancers17050791

**Published:** 2025-02-25

**Authors:** Mehdi Heidari Horestani, Golnaz Atri Roozbahani, Aria Baniahmad

**Affiliations:** Institute of Human Genetics, Jena University Hospital, 07747 Jena, Germany; mehdi.heidari.horestani@uni-jena.de (M.H.H.); golnaz.atri.roozbahani@uni-jena.de (G.A.R.)

**Keywords:** prostate cancer, histone 2A variant, H2AJ, androgen receptor, cellular senescence, SAHF

## Abstract

Prostate cancer (PCa) is a significant public health issue. The androgen receptor (AR) plays a key role in the regulation of prostate and PCa. Supraphysiological androgen levels (SALs) inhibit PCa growth and induce cellular senescence. It has been shown that H2AJ accumulates in senescent cells but the detailed mechanisms of how H2AJ is involved in androgen-controlled cellular senescence in PCa is not yet understood. H2AJ expression is enhanced in primary tumors but reduced in metastatic PCa specimens. Also, H2AJ is highly expressed in primary PrEC and C4-2 cells and much less in the metastatic LNCaP, PC3, and DU145 cell lines, which is in line with the repression of mesenchymal markers by H2AJ. Interestingly, AR is recruited to the *H2AJ* gene locus and induces its expression, suggesting that *H2AJ* is a novel direct AR target gene and part of AR signaling. Further, H2AJ controls SAL-induced cellular senescence and represses mesenchymal markers.

## 1. Introduction

Prostate cancer (PCa) is the second leading cause of cancer death in men [1]. The first line of the hormone treatment for PCa is androgen deprivation therapy (ADT) to which castration-sensitive PCa responds. ADT reduces the androgen levels to block AR signaling and inhibits the growth and survival of PCa cells [2,3]. While ADT can significantly reduce tumor burden and improve symptoms, it is not a curative treatment, and many patients eventually develop resistance to ADT, with the cancer transitioning to the castration-resistant PCa (CRPC) stage, which poses additional treatment challenges [4]. For patients with CRPC, there is an innovative treatment approach currently in clinical trials with the cycling of supraphysiological and near-castration androgen levels called Bipolar Androgen Therapy (BAT) [5]. BAT is currently being evaluated in several phase II clinical trials including TRANSFORMER [6,7,8], RESTORE [6], and COMBAT [6,7,9]. Supraphysiological androgen levels (SALs) used in BAT have been shown to inhibit PCa cell growth through various pathways including cell cycle arrest [10] and the regulation of circadian genes [11]. The inhibition of growth by SALs is associated with the induction of cellular senescence, which defines stable arrest of the cell cycle [12]. Cellular senescence is a double-edged sword that plays a crucial role in various physiological processes including development, tissue repair, and cancer prevention, but it can also contribute to aging and age-related diseases and, eventually, cancer regrowth [13]. Although AR signaling is well-established and recognized for its role in mediating cellular senescence, the specific mechanisms by which AR regulates senescence in PCa remain incompletely understood. The precise molecular pathways, key downstream effectors, and context-dependent factors influencing AR-driven senescence require further investigation.

There are several markers to detect senescent cells [14] including SA β-Gal activity staining and the p21^WAF1/Cip1^ cell cycle inhibitor [14,15]. Senescence-associated heterochromatin foci (SAHF) formation is another marker of senescent cells [16]. SAHF accumulation is associated with the stable repression of E2F target genes and usually does not occur in reversibly arrested cells [17]. On the other hand, variants of histones play roles in the induction and maintenance of cellular senescence [18] such as being induced in response to DNA damage. It was shown that H2A.X foci accumulate in persistent DNA damage and could be a trigger for cellular senescence [19]. Recently, it was shown that H2AJ accumulates in senescent cells, and its levels increase significantly during replicative and oncogene-induced senescence [20,21]. However, there is a lack of information about the role of H2AJ in AR-mediated cellular senescence in PCa.

In this study, we aimed to address whether H2AJ mediates cellular senescence and might regulate it within the SAL-induced AR signaling in PCa. Interestingly, H2AJ is expressed in C4-2 cells representing castration-resistant cells (CRPC) that were used for KD, whereas H2AJ expression was nearly undetectable in castration-sensitive (CSPC) LNCaP cells used for overexpression. In combination with ChIP-seq and RNA-seq experiments, the obtained data suggest that *H2AJ* is a direct AR target gene, being a downstream factor in AR signaling, and partially regulates SAL-induced cellular senescence, promotes the cell growth of CSPC and CRPC cells, and inhibits the expression of mesenchymal markers, which may explain the lower expression of H2AJ in PCa metastasis.

## 2. Materials and Methods

### 2.1. Cell Culture and Treatments

The LNCaP cell line, representing a castration-sensitive prostate cancer model, was acquired from Protopopov et al. [22]. The C4-2 cell line, exhibiting castration-resistant characteristics in PCa, was obtained from Thalmann et al. [23]. LNCaP cells were cultured in RPMI medium 1640 (Gibco, Life Technologies, Waltham, MA, USA) and C4-2 cells in DMEM medium (Gibco, Life Technologies, USA) as previously described [12]. Furthermore, 1 nM methyltrienolone (R1881) (Merck, Darmstadt, Germany), defined previously as SAL [12], or 0.1% dimethyl sulfoxide (DMSO) (ROTH, Dautphetal, Germany) as a solvent control was used for cell treatments after 48 or 72 h depending on the experiment.

### 2.2. siRNA and Overexpression-Vector Transfection of H2AJ

Si-mediated knockdown was performed in both cell lines by using pool ON-TARGETplus Human *H2AJ* siRNA with the “CGGCAAAGUGCGAGCAAAG”, “UGGAGUACCUUACGGCGGA”, “GCACAGACUGCUGCGCAAA”, “UAAACAAGCUGCUGGGCAA” sequences (Dharmacon, Lafayette, CO, USA) with a final concentration of 25 nM. As a negative control, pool ON-TARGETplus nontargeting control siRNA with the “UGGUUUACAUGUCGACUAA”, “UGGUUUACAUGUUGUGUGA”, “UGGUUUACAUGUUUUCUGA”, “UGGUUUACAUGUUUUCCUA” sequences (Dharmacon, USA) was used. siRNAs were transfected by the DharmaFECT reagent (Dharmacon, USA) according to the manufacturer’s protocol. For the overexpression of *H2AJ*, the cDNA sequence of *H2AJ* was purchased from Eurofins Genomics. pCDH-CMV-puro was used to insert the *H2AJ* cDNA using the jetPRIME DNA transfection kit (Polyplus, Illkirch-Graffenstaden, France) according to the manufacturer’s instructions. Briefly, 10 µg of DNA was diluted in 500 µL of jetPRIME buffer. Following several spin-down steps, 20 µL of the jetPRIME reagent was added to the mixture, and after 10 min of incubation at RT, the transfection mix was added to the plate dropwise. A day after transfection, the cells were treated with SAL or DMSO.

### 2.3. Senescence Associated β-Galactosidase Activity Staining

First, 5 × 10^4^ cells were seeded per well in 6-well plates. Then, 72 h post-treatment, the cells were washed with 1× PBS prior to fixation by 1% glutardialdehyde (ROTH, Germany) for 5 min. Staining was performed as described previously [12]. The cells were incubated at 37 °C in SA β-Gal staining solution for 24 h for C4-2 cells and 48 h for LNCaP. The staining solution contains 40 mM citric acid/sodium phosphate buffer (pH 6.0), 1 mg/mL X-gal (5-bromo-4-chloro-3-indolyl-β-D-galactopyranoside) (Invitrogen, Carlsbad, CA, USA), 5 mM K_3_Fe(CN)_6_, 5 mM K_4_Fe(CN)_6_, 150 mM NaCl, and 2 mM MgCl_2_. After the incubation step in the staining solution, the plates were washed with 1× PBS, and positively stained cells were counted using brightfield microscopy.

### 2.4. Growth Assays

First, 5 × 10^4^ cells were seeded per well in 6-well plates. Then, 72 h post-treatment, the cells were fixed and crystal violet was used to stain the cells according to a previously described protocol [24]. Excess dye was removed by washing with deionized water, and plates were dried at RT overnight. Cells were destained with Sörenson’s solution for 30 min, which contained 35 mM tri-sodium citrate, 2% HCl (37%), 40% ethanol, and water. The SPECORD^®^ 50 PLUS UV/Vis spectrophotometer (Analytik, Jena, Germany) was used to measure the absorbance at 590 nm.

### 2.5. RNA Extraction and Reverse Transcription-Quantitative Real-Time PCR (qRT-PCR)

First, 2 × 10^5^ cells were seeded per well in 6-well plates. Then, 48 h post-treatment, RNA was isolated from the cells using RNA-solv reagent (omega) according to the manufacturer’s protocol. In brief, the cell suspension was collected in a 1.5 mL Eppendorf tube. After adding RNA-solv and chloroform, the mixture was thoroughly mixed, and centrifugation was performed at 12,000× *g* for 30 min at 4 °C to achieve phase separation. RNA was precipitated by isopropanol and centrifugation at 12,000× *g* for 30 min at 4 °C. The RNA concentration was measured by the Nanodrop ND-1000 Spectrophotometer after dissolving the RNA pellet in 20 μL of DEPC-H_2_O. Then, 2 µg RNA was converted to cDNA using a High-Capacity cDNA Reverse Transcription Kit (Applied Biosystems, Waltham, MA, USA). The reaction mix for cDNA synthesis contains 4.2 μL of DEPC-H2O, 2 μL of 10X reverse transcriptase buffer, 2 μL of 10X random primers, 0.8 μL of 25X dNTP mix, and 1 μL of MultiScribe Transcriptase. To assess the transcription levels of target genes, qRT-PCR was conducted using cDNA, SsoAdvanced Universal SYBR Green Supermix (Bio-Rad, Hercules, CA, USA), and gene-specific primers in the Bio-Rad CFX96^TM^ Duet Real Time PCR machine. All primers used are listed in Table 1.

### 2.6. Protein Extraction and Western Blotting

First, 2 × 10^5^ cells were seeded per well in 6-well plates. Three days after treatment, the cells were washed once with 1× PBS, and the cell suspension was centrifuged at 2500× *g* for 5 min at 4 °C to collect the cell pellet. Then, the pellet was resuspended in 80 μL of the lysis buffer (20 mM Tris-HCl pH 8.0, 2 M NaCl, 1 mM EDTA, 1% NP-40, 1% Tergitol, 50 mM NaF, 100 μM Na_3_VO_4_, and 10 mM β-Glycerophosphate). Next, centrifugation at 12,000× *g* at 4 °C for 15 min was performed, and the supernatant containing proteins was collected. Subsequently, 30 µg of the protein extract was loaded for Western blotting. Cell lysates were separated by 12% SDS-PAGE. Afterward, proteins were transferred to PVDF membranes and incubated for 1 h in skim milk to block the membranes. Then, membranes were incubated in primary antibodies. Detection was performed by ImageQuantTM LAS 4000 (GE Healthcare Bio-Sciences AB, Uppsala, Sweden) and the quantification of bands was performed with the LabImage1D software version 7.1.0. The antibodies are listed in Table 2.

### 2.7. Chromatin Immunoprecipitation-q-PCR

First, 8 × 10^5^ C4-2 cells were seeded in 15 cm plates. Then, 72 h post-treatment, the cells were subjected to AR chromatin immunoprecipitation (ChIP) following the manufacturer’s protocol (iDeal ChIPseq Kit Diagenode, Denville, NJ, USA). In brief, 15 mL of PBS/MgCl_2_ and 60 µL of Gold ChIP cross-linking reagent were added to each plate. After 30 min of incubation at RT, cells were washed twice with PBS. Subsequently, 20 mL of PBS and 2 mL of the fixation solution were added, followed by 10 min of incubation. Afterward, 2.2 mL of glycine was added and incubated for 5 min at RT. The solution was removed and cells were washed with PBS before being collected in Falcon tubes. The cells were then centrifuged at 5000× *g* for 5 min at 4 °C. The resulting cell pellet was resuspended in 12 mL of iL1b buffer and incubated on ice for 20 min, followed by centrifugation at 5000× *g* for 5 min at 4 °C. The pellet was resuspended in 7.2 mL of iL2 buffer and centrifuged again at 5000× *g* for 5 min at 4 °C. Next, 800 µL of shearing buffer was added to the pellet, which was incubated on ice for 10 min before undergoing sonication. After sonication, the sample was centrifuged at 16,000× *g* for 10 min at 4 °C. Chromatin immunoprecipitation was performed using an AR antibody (details in Table 2), followed by de-crosslinking and DNA purification steps as per the iDeal ChIPseq Kit Diagenode protocol. Quantitative PCR was then conducted for selected regions using the specific primers listed in Table 1.

### 2.8. Immunofluorescence Staining

For immunofluorescence analysis, 15,000 cells were plated and transfected with or without si*H2AJ* in Lab-Tek chambered borosilicate cover glasses. On the following day, cells were treated with either SAL or DMSO for a duration of three days. The cells were then washed three times with 1× PBS, permeabilized using 0.2% Triton X-100 for 10 min, and washed again with 1× PBS. To block non-specific binding, cells were incubated for 1 h at RT with 5% normal goat serum (NGS) (Biozol, Hamburg, Germany). After additional washing, the nuclei were stained with DAPI solution (1 μg/mL Hoechst in 1X PBS) for 10 min. Following the final wash, the samples were mounted using Fluoromount-G^®^ (SouthernBiotech, Birmingham, AL, USA) and sealed with coverslips. Images were acquired using a Carl Zeiss LSM 880 confocal laser scanning microscope.

### 2.9. Public Data Availability

The AR ChIP-Seq dataset analyzed during the current study is available in the Gene Expression Omnibus (GEO) with accession number GSE179684 for C4-2. ChIP-seq data were analyzed for hormone-dependent recruitment of the AR to the H2AJ gene locus in C4-2 cells using IGV software version 2.13.0 (DHT, dihydrotestosterone at SAL; veh, vehicle/solvent control). hg19 was used as a reference genome visualized by IGV. GSE262744 was used for hormone-dependent recruitment of the AR to the *H2AJ* gene locus in LNCaP cells. The synthetic androgen R1881 was used at supraphysiological levels. Veh refers to the vehicle/solvent control (DMSO), while CFS refers to charcoal-stripped serum. hg38 was used as a reference genome. The expression level of *H2AJ* was compared among different cell lines by extracting the expression levels from GSE70466 for PrECs (prostate epithelial cells) and the LNCaP cell line, GSE205378 for PrSCs (prostate stromal cells), GSE172205 for C4-2, GSE141806 for 22RV1, and GSE210847 for C4-2B cell lines. The expression level of *H2AJ* for PC3 and DU145 cell lines was extracted from the Human Protein Atlas. GSE62701 Illumina bead Chip RNA expression from GEO for human fibroblast cells was used for *H2AJ* KD effects on expression levels of target genes. The senescence score list was provided by Wang et al. [25]. The expression level of *H2AJ* was analyzed between normal tissues and primary and metastatic prostate tumors from a large cohort of PCa samples using RNA-seq data from NIH genomic data common (GDC) data portal (https://portal.gdc.cancer.gov, accessed on 1 August 2024) with open access RNA-seq datasets.

### 2.10. Bioinformatics and Statistical Analyses

The PathfindR, clusterProfiler, Enrichr, and Metascape were used for pathway analyses [26,27,28,29,30]. Differentially expressed genes (DEGs) with a *p* value threshold of 0.05 were used as input for pathway analysis. Using PathFindR for pathway analysis, the *p* value was set at 0.05 and “Bonferroni” for the adjustment method. For pathways output from pathfindR, fold enrichment is a measure that indicates how much a particular pathway is overrepresented in the input gene list compared to what is expected by chance based on the background gene set. Gene–Gene correlation studies were performed using the GEPIA web tool [31]. ChIP-seq visualization was performed by the IGV software [32]. For the AR binding site analysis, JASPAR-2024 was utilized [33]. For statistical analysis, Graph Pad Prism 8.0 software was utilized. Data are expressed as the Mean ± SEM. The two-tailed student *t*-test was used for the comparison of the Mean values between two groups and two-way ANOVA with post-hoc (Tukey) test was used for multiple comparisons. The Kruskal-Wallis Test with post-hoc Dunn’s multiple comparisons test was used in R for statistical analysis for experiments with 2 biological replicates. The Z score of normalized counts was calculated to standardize the expression levels and as a suitable option to directly compare the expression of *H2AJ* from different datasets within a common scale. Positive Z-scores indicate above-average expression levels relative to the dataset, while negative Z-scores represent below-average expression levels. A *p* value of less than 0.05 was considered statistically significant, with significance levels denoted as follows: * *p* < 0.05; ** *p* < 0.01; *** *p* < 0.001; **** *p* < 0.0001; ns: not significant.

## 3. Results

### 3.1. H2AJ Is a Direct Target of AR and Specifically Is Expressed in C4-2 Cells

H2AJ with UniProt ID “Q9BTM1”, is a variant of the canonical histone H2A with only 5 amino acids difference [20]. It has been shown that H2AJ is expressed at low levels in proliferative human fibroblasts and strongly accumulates in chromatin of senescent human fibroblasts with persistent DNA damage [20]. Here, pathway analysis of our recently published SAL transcriptome [11] revealed that H2AJ is part of many cellular senescence and chromatin organization-related pathways (Figure 1A). The expression level of this histone variant was compared within a large cohort of prostatic tissue samples between normal tissues (52 samples), primary (539 samples), and metastatic prostate tumors (100 samples) using RNA-seq data from NIH genomic data common (GDC) data portal with open access RNA-seq datasets. Although the expression of *H2AJ* transcript is significantly upregulated in primary tumors compared to normal tissues, interestingly, the level of *H2AJ* is drastically reduced in the metastatic samples (Figure 1B). The survival plot of PCa patients indicates that a higher level of *H2AJ* is significantly associated with better patient survival and lower expression to worse outcomes, which may link low H2AJ expression to the observed metastatic prostate tumors (Figure 1C). Therefore, our hypotheses were that, on the one hand, H2AJ promotes the growth of PCa cells, whereas, on the other hand, H2AJ inhibits the expression of mesenchymal markers as a pathway required for metastasis [34]. To obtain some hints for our hypotheses, we compared the expression of *H2AJ* in primary prostate epithelial cells (PrEC) with primary stromal cells (PrSC). The data suggest a lower expression of *H2AJ* in epithelial cells compared to stromal cells (Figure 1D). Further, analyzing PCa cell lines derived from metastasis suggests a lower expression of *H2AJ* in the lymph node metastasis LNCaP (Figure 1D and Appendix A), the bone metastasis PC3 and C4-2B, and brain metastasis-derived DU145 [35,36,37,38], as well as in androgen-sensitive non-metastatic 22RV1 cells [39] (Figure 1D), whereas higher expression of *H2AJ* was observed in the non-metastasized xenograft C4-2 tumors [23] (Figure 1D and Appendix A).

To analyze the androgen regulation of *H2AJ* expression, our transcriptome data from both LNCaP and C4-2 cells were used [10,11]. The data suggest that H2AJ expression is astonishingly only detectable in C4-2 cells. The pronounced expression differences in *H2AJ* mRNA between LNCaP and C4-2 cells were confirmed by qRT-PCR (Appendix A). In line with this, the H2AJ protein levels exhibit strong differences between LNCaP and C4-2 cells (Figure 1B).

Of note, RNA-seq datasets indicate the upregulation of *H2AJ* expression by SALs, which was validated by qRT-PCR and Western blotting at both mRNA and protein levels (Figure 1E,F and Appendix A), but no regulation of *H2AJ* was observed in C4-2 cells treated with Enzalutamide (Enz) as a second-generation AR inhibitor (Appendix A), suggesting that H2AJ is part of the signaling mediated by SALs.

AR recruitment to the *H2AJ* gene locus was analyzed using chromatin-immunoprecipitation (ChIP)-seq using an AR antibody with and without SAL treatment in C4-2 and LNCaP cells [40,41]. Several chromatin recruitment sites of AR were identified at the *H2AJ* gene locus that are induced by SALs in C4-2 cells (Figure 1G). In contrast, only one AR recruitment site to the *H2AJ* gene was detected in LNCaP cells, with this being recruited to a different genomic locus compared to C4-2 cells (Appendix A). For validation, ChIP-qPCR was conducted to confirm AR binding sites within the *H2AJ* gene locus identified within the ChIP-seq datasets. Primers were designed for selected peaks and genomic qPCR of the ChIP material was performed. In agreement with ChIP-seq data, the recruitment of AR to the gene locus of *H2AJ* was confirmed by ChIP-qPCR (Figure 1H), suggesting that *H2AJ* is a direct AR target, and its expression is also regulated by SALs as the recruitment of AR to the peak1 region was further induced by SALs (Figure 1H).

The AR is a transcription factor that binds to specific DNA sequences called androgen response elements (AREs) [42]. Classical AREs have a consensus palindromic sequence AGAACAnnnTGTTCT, with AGAACA being a half site. Variations in ARE sequences can affect AR binding affinity and transcriptional activity [42]. In addition to ChIP-qPCR, in silico AR motif binding site analysis was performed using JASPAR-2024 [33] to validate the possibility of AR binding within the *H2AJ* gene locus. The sequences of peaks shown in Figure 1G (green box), along with a region lacking any peaks (as a negative control, red box), were used as input for motif analysis. As presented in Table 3, all three peaks harbor ARE half-site motifs with higher scores compared to the negative control. This finding aligns with the ChIP-seq and ChIP-qPCR data, suggesting that these closely positioned multiple binding sites likely form homotypic clusters [43] and indicating that these peaks together may enhance binding affinity and be essential for the full functionality of AR. Furthermore, the number of AR binding sites appears to influence the expression level of the *H2AJ* gene, consistent with reports that the quantity and arrangement of binding sites can regulate gene expression levels [44]. Based on this information, these binding sites may collectively be critical for full AR functionality and the regulation of target gene expression. Notably, in LNCaP cells, only one binding site was identified, which can explain the very low expression level of *H2AJ* in this cell line.

These data suggest that *H2AJ* is a novel, direct, positive target gene of AR, and the different recruitment of AR to the *H2AJ* gene locus might explain the differences in expression and regulation of *H2AJ* within these two cell lines.

To address the question of whether H2AJ can, within a transcriptional feedback loop, regulate the expression of AR, the change in the protein level of AR was analyzed using small interference (si)-mediated KD of *H2AJ*. In addition, to analyze a possible change in the transcriptional activity of AR mediated by H2AJ, the mRNA levels of several direct AR target genes were analyzed using si*H2AJ* knockdown. The knockdown efficiency was validated at both mRNA and protein levels (Figure 1E,F).

The data suggest that the AR protein level and the expression levels of known direct AR target genes were not significantly affected by the *H2AJ* KD (Figure 1B and Appendix A). This suggests that H2AJ does not measurably control either the AR protein level or transcriptional activity, thus indicating that H2AJ is downstream of AR signaling and suggesting the identification of a novel AR signaling pathway.

In line with this, gene correlation studies using datasets of prostate adenocarcinoma (PRAD) from TCGA in GEPIA revealed a positive correlation between *H2AJ* expression and each of the analyzed AR direct target genes (Appendix A). This indicates that increased AR signaling is associated with increased *H2AJ* expression in PCa.

### 3.2. H2AJ KD Reduces Growth and Induces Cellular Senescence Through the p21^WAF1/Cip1^ Cyclin-Dependent Kinase Inhibitor

Based on the appearance of many pathways linking the *H2AJ* gene to cellular senescence (Figure 1A), we hypothesized that H2AJ may partially control SAL-induced cellular senescence. To address this hypothesis, si-mediated KD of *H2AJ* was employed. The level of KD was confirmed at both mRNA and protein levels (Figure 1E,F). The senescence-associated beta-galactosidase (SA β-Gal) activity staining suggests an induction of cellular senescence by the KD, indicating that H2AJ regulates cellular senescence. In the presence of SAL, a further but less pronounced increase in SA β-Gal positively stained cells was detected (Figure 2A,E). However, in LNCaP cells with low endogenous H2AJ levels, no detectable changes in cellular senescence or cell growth were observed 72 h post-treatment (Appendix A).

At the biochemical level, the expression level of p21^WAF1/Cip1^, as a marker of cellular senescence [11], was analyzed and compared to the *H2AJ* KD samples. The KD of *H2AJ* upregulates p21^WAF1/Cip1^ at both the protein and mRNA levels (Figure 1B and Figure 2B), suggesting that *H2AJ* KD-induced cellular senescence is associated with an increase in p21^WAF1/Cip1^. In LNCaP cells, the mRNA levels of *CDKN1A* showed no significant differences between *H2AJ* KD samples and control samples, with or without SAL treatment (Appendix A), confirming the observations shown in Appendix A.

Furthermore, the expression of *E2F1*, as a critical cell cycle-promoting factor [45], was analyzed. The data indicate that SAL reduces *E2F1* expression levels, confirming reduced growth by SAL (Figure 2C). A further significant reduction in *E2F1* is detectable after the KD of *H2AJ* (Figure 2C). Therefore, we speculated that in C4-2 cells *H2AJ* KD will result in reduced cell growth. In line with this, the growth curve analysis from 72 h and up to 6 days of treatment further confirmed the inhibitory growth effect of the *H2AJ* KD (Figure 2D–F) and suggests that H2AJ promotes growth.

### 3.3. H2AJ KD Increases the Formation of Senescence Associated Heterochromatin Foci (SAHF) and Expression of Mesenchymal Markers

SAHFs are microscopically discernible condensed heterochromatic structures and have been used as a senescence marker in human cells [46]. SAHF structures lead to transcriptional repression of proliferation-promoting genes and help the maintenance of the senescence state. It has been shown that SAL increases the SAHF formation [12]. Since histones play a crucial role in chromatin assembly, DAPI staining was performed with C4-2 cells treated with and without SAL in combination with or without siRNA-mediated KD of *H2AJ* in order to analyze whether H2AJ is involved in SAHF formation. The data suggest that SAL increases the level of SAHF, which is in agreement with previously published data [12], and that SAHF levels are further enhanced via KD of *H2AJ* (Figure 3A and Appendix A). These data suggest that the reduction in H2AJ enhances SAHF formation as a further induction that H2AJ regulates cellular senescence in PCa cells.

As the expression of H2AJ is strongly reduced in metastatic samples of a large cohort of PCa patients (Figure 1B), we hypothesized that H2AJ plays a role in the regulation of epithelial–mesenchymal transition (EMT) markers. To analyze this hypothesis, the mRNA levels of *CDH1*-encoding E-cadherin, *CDH2*-encoding N-cadherin, and *Vim*-encoding Vimentin were investigated. Although no significant regulation of the epithelial marker *CDH1* was detected in either SAL-treated or *H2AJ* KD samples, the expression levels of both mesenchymal markers *CDH2* and *VIM* were increased by the KD of *H2AJ* (Figure 3B–D and Appendix A), suggesting that H2AJ inhibits the expression of mesenchymal markers and aligns with the reduced levels of *H2AJ* mRNA observed for the metastatic samples of a large cohort of PCa patients (Figure 1B).

In agreement with the functional data, gene correlation analyses for H2AJ with EMT factors were performed using the GEPIA web tool for prostate adenocarcinoma (PRAD). The Pearson correlation coefficient calculation for the PRAD datasets derived from TCGA (The Cancer Genome Atlas Program) from the National Cancer Institute, USA, was used. The data suggest a positive and significant correlation for epithelial markers with H2AJ expression and a significant negative correlation for mesenchymal markers (Figure 3E–G). In addition, a specificity control correlation analysis was performed with H2AX, another variant of H2A that is a known marker of DNA damage-induced senescent cells [47], as well as histone H2AFZ, which is involved in the progression of PCa [48], with each of the EMT markers. In contrast to H2AJ, the results show only a very feeble correlation between H2AX or H2AFZ with each of the EMT markers. This suggests a specific and significant correlation between the H2AJ histone variant and EMT markers (Appendix A).

### 3.4. Overexpression of H2AJ in LNCaP Cell Line Reduces Cellular Senescence and Induces Cell Growth

Since LNCaP cells express no or very low levels of H2AJ, we analyzed the role of H2AJ in mediating cellular senescence by overexpression (OE-*H2AJ*) in these cells. As expected, the overexpression of *H2AJ* in LNCaP cells did not show a measurable effect on AR protein levels (Figure 4A), with this being in line with the results of C4-2 cells using KD experiments. Upon analyzing the SAL-mediated induction of both protein and mRNA levels of p21^WAF1/Cip1^, the data suggest that the overexpression of *H2AJ* reversed this induction to basal levels (Figure 4A,B). This suggests that the overexpression of *H2AJ* has a rescue effect induced by SAL.

In agreement with this, the *E2F1* mRNA level, as a cell cycle-promoting factor, was significantly increased by OE-*H2AJ* also in the presence of SAL (Figure 4C). In line with this, our functional analysis suggested that SAL-induced cellular senescence is reversed to basal levels after *H2AJ* overexpression (Figure 4D,E), confirming the Western blot results of reduced p21^WAF1/Cip1^ protein levels comparing the fold induction by SAL with and without OE-*H2AJ*. This further supports the notion that H2AJ is part of AR signaling and may counteract the induction of cellular senescence as a feedback mechanism.

To confirm whether the senescence levels correlate with cell growth, treatments of 72 h and up to 6 days were analyzed. The results indicate a significant growth enhancement in *H2AJ*-overexpressed samples compared to the control empty vector (EV) (Figure 4F,G). These findings align with the KD data and highlight the role of H2AJ in regulating cellular senescence and growth by SAL treatment. In line with the KD of *H2AJ* in C4-2 cells, the overexpression of *H2AJ* in LNCaP cells had the opposite effect on the regulation of EMT factors by exhibiting a significant downregulation of mesenchymal factors (Appendix A). This indicates that H2AJ represses the expression of mesenchymal factors.

Thus, these data suggest that this histone variant inhibits the expression of mesenchymal markers and counteracts the androgen-induced cellular senescence.

### 3.5. Bioinformatic Analyses Revealed a Large Overlap of H2AJ Transcriptome with the Cellular Senescence Score of PCa

The Venn diagram depicts the overlap genes between the cellular senescence score of PCa and the *H2AJ* KD transcriptome dataset (Figure 5A). The cellular senescence score in PCa is a set of analyzed genes identified by machine-learning models and suggested to be involved in cellular senescence [25]. In total, 1098 out of the 1254 senescence score genes of PCa show commonality with the H2AJ transcriptome, which strongly emphasizes the role of H2AJ in the regulation of cellular senescence. As the first step, pathway analysis was performed for the transcriptome data of *H2AJ* KD samples for significantly expressed genes, with the cutoff *p* value of 0.05 including 1180 genes. Many statistically significant cellular senescence-related pathways emerged including “Formation of Senescence-Associated Heterochromatin Foci (SAHF)”, “Cellular Senescence”, and circadian clock and metastasis-related pathways (Table 4 and Figure 5B). The pathway analysis results are in agreement with the identified activity of H2AJ in mediating cellular senescence and are part of SAHF formation.

Furthermore, to assess how the H2AJ transcriptome is associated with the cellular senescence score in PCa [25], DEGs from the *H2AJ* KD transcriptome with cutoff *p* values of 0.05 were compared with the senescence score gene list with the cutoff *p* value of 0.05. As expected, pathway analysis of these common factors after the cutoff *p* value of 0.05 also revealed pathways linked to chromatin, cell cycle regulation, and DNA repair (Figure 5C). Among the 1098 common genes, 40 were both upregulated by *H2AJ* KD and downregulated in the cellular senescence score of PCa, while 24 genes were both downregulated by *H2AJ* KD and upregulated in the senescence score list (Figure 5D). The pathways for these 64 differently regulated common factors are depicted in Appendix A. Interestingly, in addition to the significant pathways linked to mechanisms involved in cell membrane structure, cell signaling, cell-to-cell communication, and cellular homeostasis, several significant pathways emerged that are related to cellular senescence and cell cycle regulation and were also identified among pathways for differently regulated common factors.

Thus, the data suggest the identification of a novel AR-histone pathway in that the histone variant H2AJ regulates cellular senescence as a direct AR target gene within the androgen signaling mediated by SAL. Further, the data suggest that H2AJ promotes growth as a possible feedback loop to SAL-mediated growth inhibition and that H2AJ inhibits the expression of mesenchymal markers and may therefore be downregulated during PCa metastasis.

## 4. Discussion

AR signaling plays a central role in PCa regulation, exerting its effects through both genomic and non-genomic pathways [49]. Genomic AR signaling involves the direct binding of AR to AREs in the DNA, regulating the transcription of target genes that influence cancer progression. In contrast, non-genomic AR signaling is characterized by rapid, cytoplasm-mediated interactions with signaling proteins such as Src and Akt, leading to phosphorylation cascades that affect various cellular processes like cellular senescence [10,11,50]. There are also some other known AR signaling pathways mediating cellular senescence in PCa, including p53-p21 [51], p16-pRb-E2F1 [12], BHLHE40-CCNG2 [11], LYL1-p27Kip1 [52], and AR-*MIR503HG*-miR-424-5p signaling [53] axes that collectively lead to cell cycle arrest and senescence regulation. Despite these well-established mechanisms of AR signaling and the knowledge that AR signaling mediates cellular senescence, there still remains a significant gap in understanding how AR mediates cellular senescence in PCa.

Histone variants play a crucial role in various cellular processes. H2AX is a histone H2A variant that is swiftly phosphorylated at serine 139 upon induction by double-strand breaks (DSBs), leading to the formation of γ-H2AX. This phosphorylation is a pivotal event of the DNA damage response [54], with elevated γ-H2AX levels being linked to cellular senescence [55]. On the other hand, H2AFZ is involved in the progression of PCa, especially during the shift to androgen independence. Its deregulation promotes oncogenic gene expression driving cell proliferation and aiding cancer progression [48]. Additionally, H2AFZ displays an altered genomic distribution in PCa, showing increases at transcription start sites (TSS) of oncogenes and decreases at those of tumor-suppressor genes [48,56]. However, information concerning the functions of H2AJ in cancer cells requires further exploration.

Histone variant H2AJ differs from the canonical H2A by only the “A11V” substitution and the presence of the “SQK” motif in its C-terminal. Interestingly, H2AJ is specific to mammals [20]. The canonical H2A is expressed mostly in the S phase of the cell cycle, while H2AJ is expressed constitutively throughout the cell cycle. This variant is the only histone, which is located on chromosome 12, whereas the other histones are clustered on chromosomes 1 and 6 [20]. It was shown that H2AJ accumulates in human fibroblast senescent cells, particularly by experiencing persistent DNA damage. This accumulation is associated with the expression of several inflammatory genes that contribute to the senescent-associated secretory phenotype (SASP) [20,21]. Conversely, following ionization radiation exposure, cells with higher levels of H2AJ exhibit a weaker induction of the cellular senescence marker p21^WAF1/Cip1^, which supports the assertion that higher levels of H2AJ correlate with reduced cellular senescence induction in response to ionization radiation [46]. This information suggests that H2AJ accumulation is linked to both the reduction in senescence and the induction of senescence-associated inflammatory responses. In the early stages following DNA damage, increased levels of H2AJ may help mitigate the induction of senescence, but over time, as cells accumulate damage and enter a senescent state, H2AJ levels rise and contribute to the inflammatory response characteristic of SASP, suggesting that the role of H2AJ can vary depending on the cellular context and timing.

Focusing on cancer, elevated H2AJ expression has been detected in several carcinoma types, especially in luminal breast and PCa, which indicates that H2AJ may act as a potential biomarker for identifying these cancer subtypes [57]. In PCa, both AR antagonists and agonists at supraphysiological levels can induce cellular senescence [50]. AR antagonists, which inhibit AR-mediated transactivation, can also induce cellular senescence, suggesting that not all AR functions are blocked by antagonists and that this process does not involve classical genomic AR signaling [50]. Notably, SAL treatment enhances H2AJ expression while Enz as an AR antagonist does not have a significant effect on H2AJ level. Interestingly, we could not detect the expression of H2AJ in the CSPC LNCaP cells, which offered the possibility to overexpress *H2AJ* in these cells to analyze cellular senescence. On the other hand, C4-2 cells that express H2AJ were used for KD experiments. Surprisingly, in C4-2 cells, the KD of *H2AJ* led to a strong induction of cellular senescence in accordance with cell growth inhibition. The overexpression of *H2AJ* in LNCaP cells revealed the opposite results to the KD of *H2AJ* in C4-2 cells by enhancing cell growth and reducing SAL-mediated cellular senescence, suggesting a role of H2AJ in AR signaling and the regulation of SAL-induced cellular senescence in PCa. Our data further suggest that H2AJ is likely a factor in non-metastasized PCa cells and might be a repressor for the mesenchymal transition. Furthermore, our data suggest that AR upon SAL treatment can be recruited to the *H2AJ* gene.

The drastic downregulation of H2AJ in metastatic PCa strongly suggests an important role of H2AJ in the regulation and inhibition of metastasis. In line with this, correlation analysis for EMT-regulating factors with *H2AJ* showed a high co-expression correlation between *H2AJ* and EMT-regulating factors. Our data suggest a functional link to the expression of mesenchymal markers that are regulated by H2AJ. Previously, we have shown that androgens induce a distinct response of EMT factors in PCa cells [58]. Thereby, androgen treatment upregulates the expression of *Vim* specifically in LNCaP cells but, surprisingly, not in C4-2 cells [58]. In contrast, the EMT regulators slug and snail, ZEB1 and ZEB2, are controlled by androgens in both LNCaP and C4-2 cells [58]. The KD of *H2AJ* in C4-2 cells indicates that mesenchymal markers *Vim* and *CDH2* expressions are enhanced at the SAL level, and OE-*H2AJ* in LNCaP cells inhibits these factors, which might be explained by the specific differential expression of H2AJ. These data together suggest that H2AJ controls the expression of mesenchymal marker expression in PCa cells.

It has been shown that histone variants and modifications play crucial roles in the formation of SAHF, which is an important structure in the process of cellular senescence [17,59]. Variants of the H2A histone such as H2AJ play a significant role in chromatin organization. H2AJ has been linked to reduced SAHF formation [21], while in another study, the accumulation of H2AJ was shown to correlate with the formation of SAHF in fibroblast cells undergoing ionization radiation-induced premature senescence [46]. This suggests that while reduced H2AJ levels may induce SAHF formation, its expression also promotes SAHF, indicating a complex regulatory role for H2AJ in the chromatin dynamics associated with cellular senescence.

## 5. Conclusions

Taken together, the data from this study suggest that H2AJ is an inhibitor of SAL-mediated cellular senescence, an inhibitor of the expression of mesenchymal markers, and a promoter of PCa cell growth. Interestingly, H2AJ is specifically highly expressed in primary PrEC and C4-2 PCa cells and very low in metastatic LNCaP, PC3, and DU145 cells, which is in line with the repression of mesenchymal markers by H2AJ. H2AJ expression is induced by androgens, which is in line with enhanced AR recruitment to chromatin by SAL, suggesting that H2AJ is part of AR signaling and partially controls SAL-induced cellular senescence and the distinct response of the expression of mesenchymal markers by AR.

## Figures and Tables

**Figure 1 cancers-17-00791-f001:**
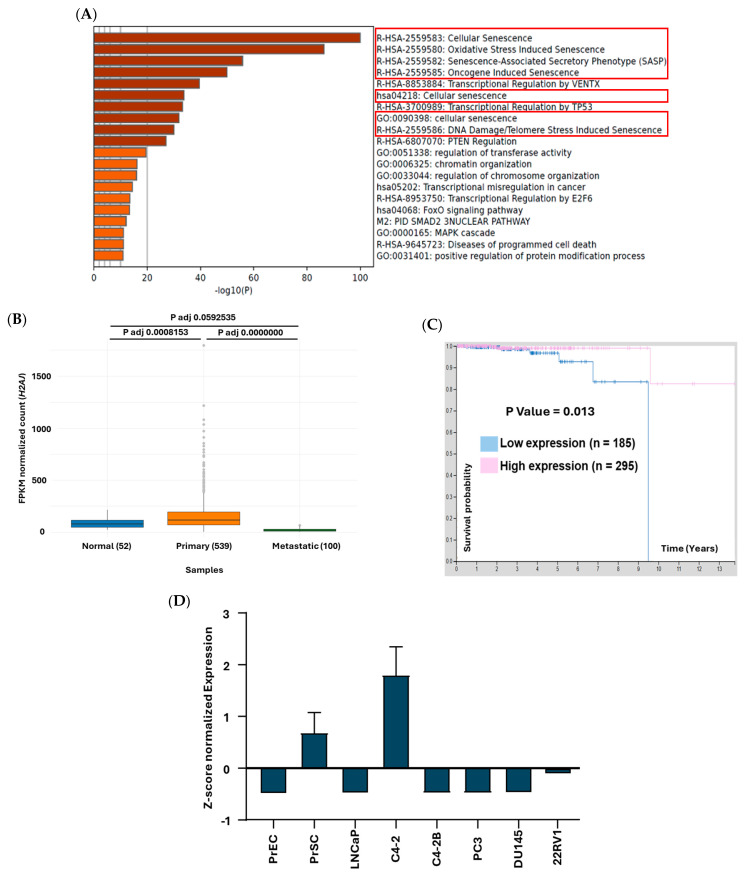
H2AJ expression is induced by SAL in C4-2 cells. (**A**) Pathways of our SAL transcriptome data that contain the H2AJ gene were analyzed using Metascape. (**B**) The expression level analysis of *H2AJ* in normal samples and primary and metastatic PCa tumors of a large patient cohort using RNA-seq data from NIH genomic data common (GDC) data portal; 52 normal, 539 primary, and 100 metastatic samples are included in this analysis. Normalized counts are indicated as fragments per kilo base pair transcript per million reads (FPKM). (**C**) Survival plot retrieved from The Human Protein Atlas indicating the significant association between expression levels of *H2AJ* and survival probability. The *H2AJ* expression cut-off is 689.74 FPKM normalized counts. “n” indicates the number of samples. (**D**) The expression levels of *H2AJ* across various PCa cell lines were extracted from different GEO datasets and the Human Protein Atlas and compared. Z-score normalization was performed to standardize the expressions from different datasets for direct comparison of the expression levels. Z-score normalized gene expression data revealed comparable trends across datasets despite differences in the original normalization methods. Positive Z-scores indicate above-average expression levels relative to the dataset, while negative Z-scores represent below-average expression levels. The Z-score reflects relative expression within the dataset and not absolute expression levels. (**E**) Western blot data from si-control (siCON) and *H2AJ* knockdown (si*H2AJ*) samples with and without SAL treatment (*n* = 3). β-Actin served as loading control. The numbers indicate the band intensities normalized to β-Actin and relative to siCON DMSO. The size of AR is 110 KDa, H2AJ is 14 KDa, p21^WAF1/Cip1^ is 21 KDa, and β-Actin is 43 KDa. (**F**) Detection of *H2AJ* mRNA levels by SAL treatment and KD efficiency analyzed by qRT-PCR in C4-2 cell line (*n* = 3). The mRNA levels of both housekeeping genes *α-Tubulin* and *TBP* were used for normalization of expression levels. (**G**) ChIP-seq data were analyzed for hormone-dependent recruitment of the AR to the *H2AJ* gene locus in C4-2 cells using IGV software (DHT, dihydrotestosterone at SAL; veh, vehicle/solvent control). hg19 was used as a reference genome visualized by IGV. Green and red boxes were selected for AR motif-binding site analysis via JASPAR-2024. (**H**) Validations for peaks 1 and 2 of AR recruitments to the *H2AJ* gene locus from ChIP-seq labeled in Figure 1G by performing ChIP-qPCR in C4-2 cells. *p* value < 0.001 = ***, ns = non-significant.

**Figure 2 cancers-17-00791-f002:**
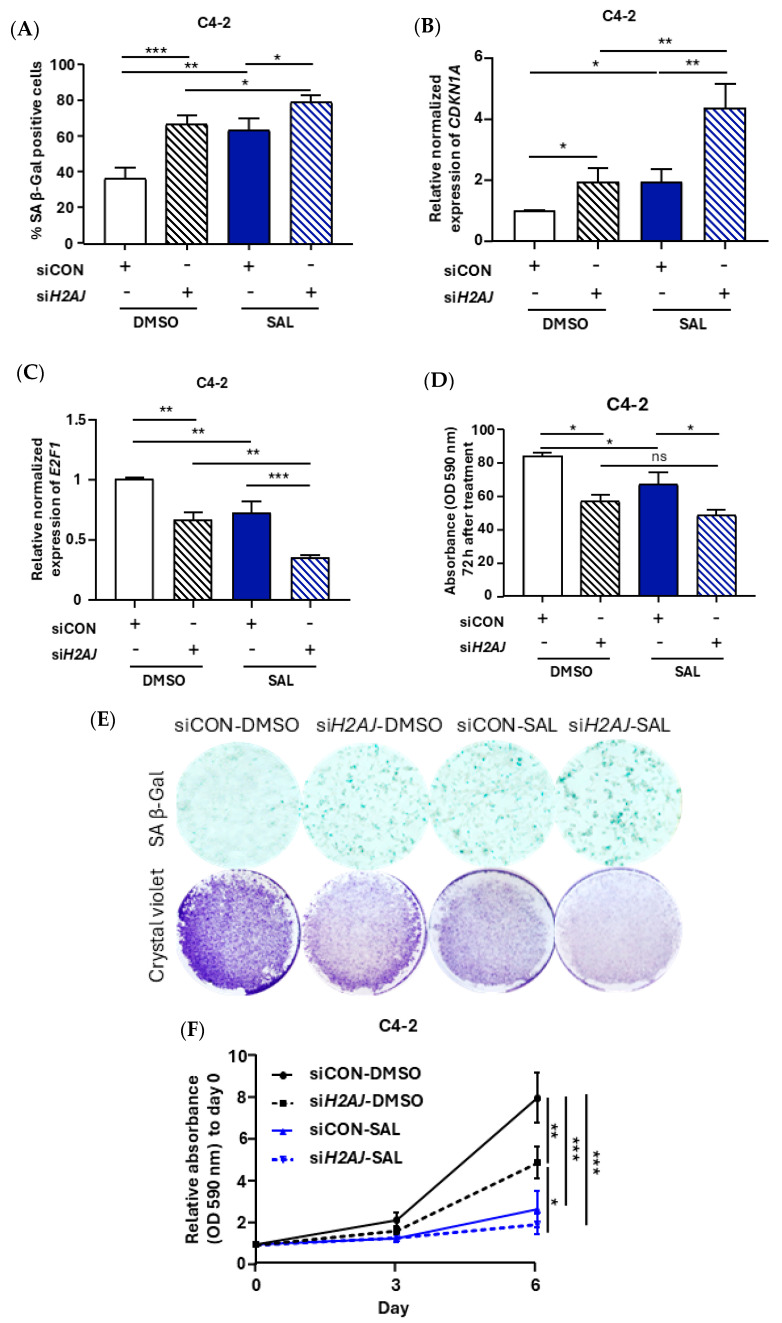
The KD of *H2AJ* induces cellular senescence and reduces cell growth. (**A**) The expression level of *CDKN1A* mRNA encoding p21^WAF1/Cip1^ in si-control (siCON) and *H2AJ* KD (si*H2AJ*) C4-2 cells (n = 3). (**B**) The mRNA level of *E2F1* in siCON samples as well as in *H2AJ* KD C4-2 cells (*n* = 3). (**C**) SA β-Gal activity staining for siCON and *H2AJ* KD samples (*n* = 3). (**D**) Quantification of crystal violet absorbance 3 days after treatment to analyze the growth of SAL-treated and *H2AJ* KD with and without SAL samples (*n* = 3). (**E**) Representative pictures of SA β-Gal activity- (upper panel) and crystal violet-stained cells (lower panel) (**F**) Growth curve analyzed by crystal violet for up to 6 days of control cells and samples with the KD of *H2AJ* in C4-2 cells (*n* = 2 biological replicates, each contains 2 technical replicates). *p* value <0.001 = ***, <0.01 = **, <0.05 = *, ns = non-significant.

**Figure 3 cancers-17-00791-f003:**
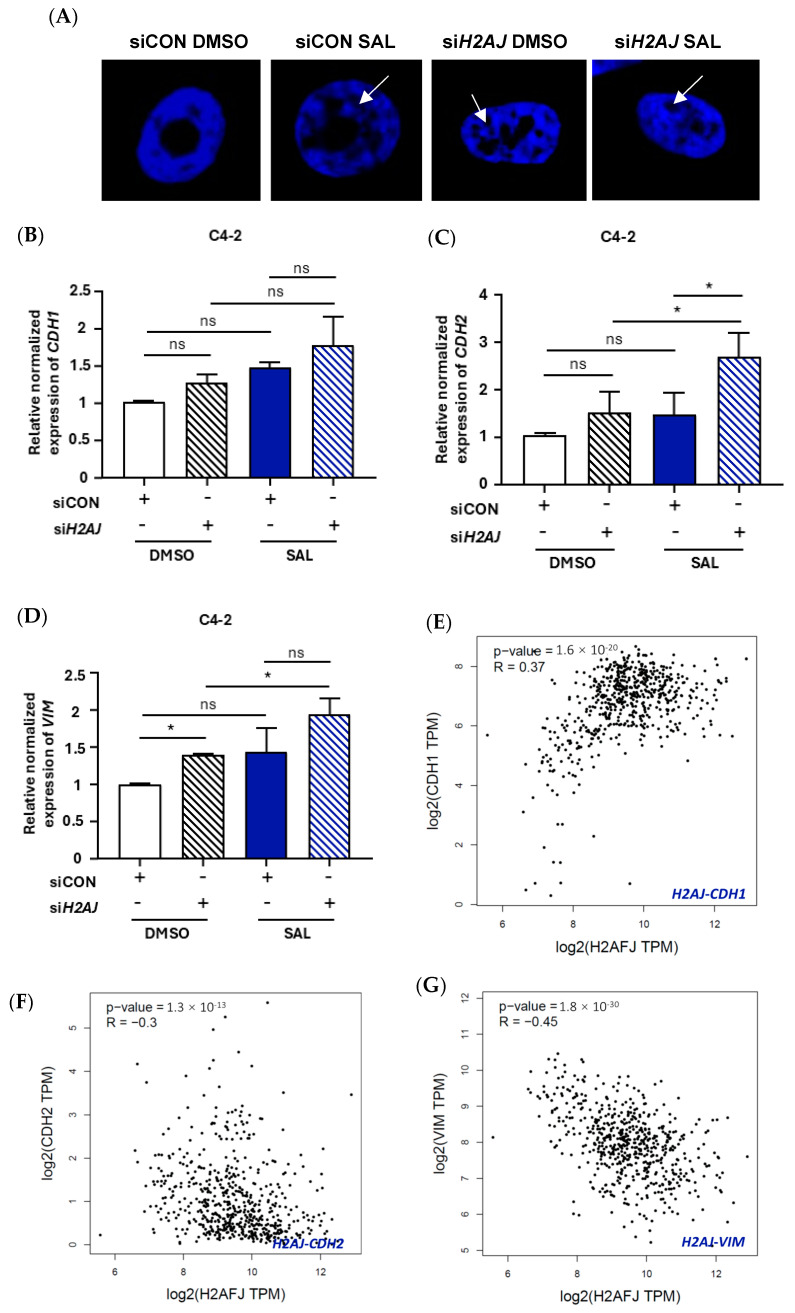
H2AJ regulates SAHF formation and the expression of mesenchymal markers. (**A**) DAPI staining of C4-2 cells treated with SAL or DMSO as solvent control in combination with and without KD of *H2AJ* (siCON and siH2AJ, respectively). Arrows point to SAHFs (*n* = 2 biological replicates with each containing 5 technical replicates). The white bars correspond to 10 µm scale. (**B**–**D**) The mRNA levels of *CDH1*, *CHD2*, and *Vim*, respectively, were detected by qRT-PCR with and without KD of *H2AJ* (*n* = 3). (**E**) Gene–Gene expression correlation between *H2AJ* and *CDH1* using PRAD datasets retrieved from GEPIA. (**F**) Gene–Gene expression correlation between *H2AJ* and *CDH2*. (**G**) Gene–Gene expression correlation between *H2AJ* and *Vim*. *p* value <0.05 = *, ns = non-significant.

**Figure 4 cancers-17-00791-f004:**
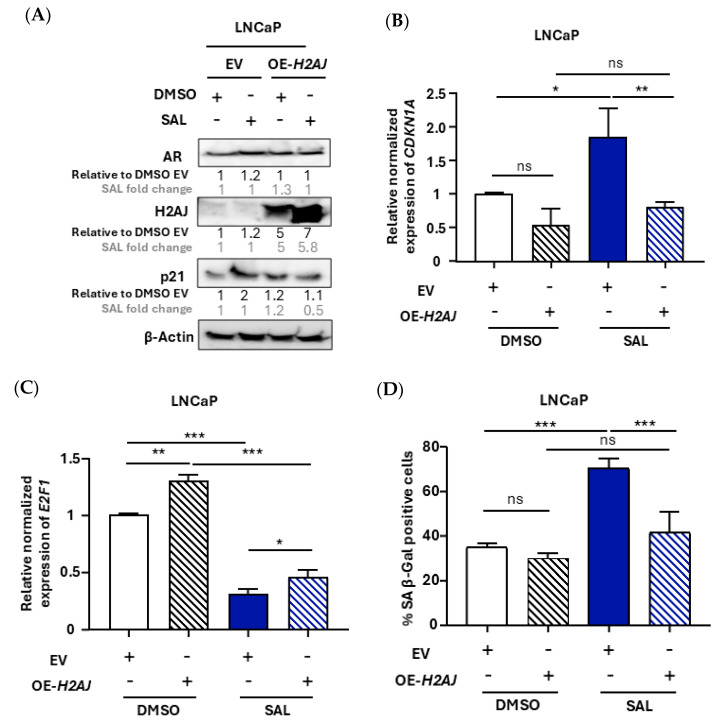
Overexpression of *H2AJ* reduces cellular senescence and induces growth in LNCaP cells. (**A**) Western blot of treated LNCaP cells to detect p21^WAF1/Cip1^ with *H2AJ* overexpression (OE-*H2AJ*) or empty vector (EV) with and without SAL treatment. DMSO served as solvent control (*n* = 3). The size of AR is 110 KDa, H2AJ is 14 KDa, p21^WAF1/Cip1^ is 21 KDa, and β-Actin is 43 KDa. (**B**) *CDKN1A* mRNA-encoding p21^WAF1/Cip1^ levels were measured by qRT-PCR in LNCaP cells with and without *H2AJ* overexpression (*n* = 3). (**C**) *E2F1* mRNA levels were analyzed by qRT-PCR in LNCaP cells (*n* = 3) (**D**) Quantification of SA β-Gal activity staining (*n* = 3). (**E**) Representative pictures of SA β-Gal activity staining and growth analyzed by crystal violet staining. (**F**) Quantification of crystal violet staining 3 days after treatments (*n* = 3). (**G**) LNCaP growth curve analyzed by crystal violet staining with and without overexpression of *H2AJ* (OE-H2AJ) and with or without SAL treatment (*n* = 2, each contains 2 technical replicates). *p* value <0.001 = ***, <0.01 = **, <0.05 = *, ns = non-significant.

**Figure 5 cancers-17-00791-f005:**
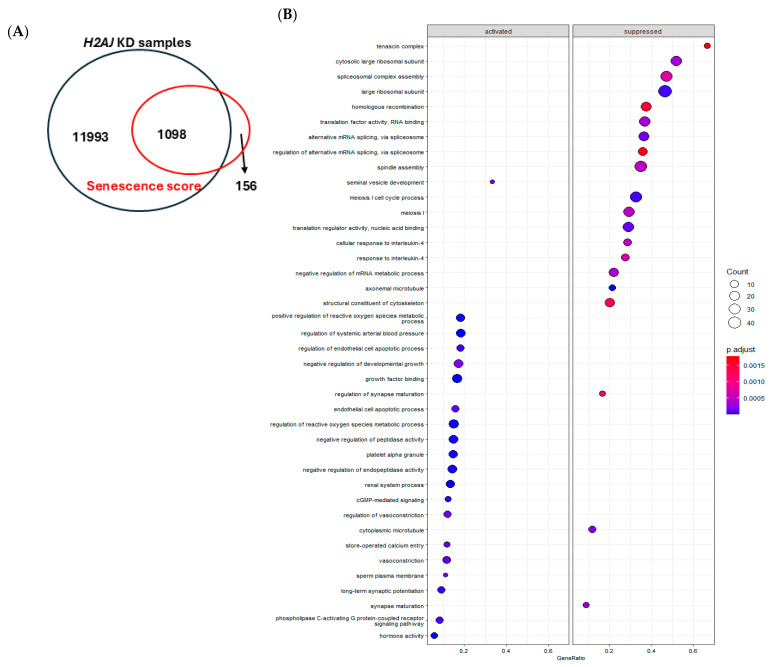
*H2AJ* transcriptome data show strong overlap with cellular senescence score of PCa. (**A**) Venn diagram depicts the overlap of genes between the *H2AJ* KD and the PCa senescence score. (**B**) Pathway analysis for identification of significant genes from *H2AJ* KD samples was performed by clusterProfiler. (**C**) Pathway analysis for common significant factors between *H2AJ* KD and the cellular senescence score of PCa performed by clusterProfiler. (**D**) List of common genes that are differently regulated between *H2AJ* KD and the cellular senescence score; 40 of these genes are upregulated in *H2AJ* KD samples and downregulated in the senescence score list, while 24 of genes are downregulated in the *H2AJ* KD samples and upregulated in the senescence score list.

**Table 1 cancers-17-00791-t001:** Sequence of primers used in qRT-PCR (5′…3′).

*KLK3*	FRW: GAGGCTGGGAGTGCGAGAAG REV: TTGTTCCTGATGCAGTGGGC
*TMPRSS2*	FRW: CCTGCAAGGACATGGGCTATA REV: CCGGCACTTGTGTTCAGTTTC
*E2F1*	FWD: GCAGAGCAGATGGTTATGG REV: GATCTGAAAGTTCTCCGAAGAG
*NKX3-1*	FWD: CCGAGACGCTGGCAGAGACC REV: GCTTAGGGGTTTGGGGAAG
*H2AJ*	FWD: AGCGGTTTGTCTCCGTCTCTC REV: CTCCGCCGTAAGGTACTCCAA
*CDKN1A*	FWD: GCAGACCAGCATGACAGATTTC REV: AGAAGATGTAGAGCGGGCCT
*CDH1*	FWD: GGCTGGACCGAGAGAGTTTC REV: TGCTGTTGTGCTTAACCCCT
*CDH2*	FWD: AGGGATCAAAGCCTGGAACAT REV: CTTGGAGCCTGAGACACGAT
*VIM*	FWD: TGGCACGTCTTGACCTTGAA REV: AGCTCCTGGATTTCCTCTTCG
*TBP*	FRW: GATCTTTGCAGTGACCCAGCATCA REV: CTCCAGCACACTCTTCTCAGC
*TUBA*	FRW: TGGAACCCACAGTCATTGATGA REV: TGATCTCCTTGCCAATGGTGTA
*H2AJ-ChIP1*	FRW: TGGACTCATTTACAAAGAGACCCGG REV: TTGTCTTTGAGGCTATGGGCC
*H2AJ-ChIP2*	FRW: GTGAACGATTGGCTGGCTGTG REV: GGGTTGTCCGTAGACGTTGCTATG
*H2AJ-ChIP-Negative*	FRW: CTGGAGTTGTAGGCGAGAGGTG REV: TGCCCAGCAGCTTGTTTAACTC

**Table 2 cancers-17-00791-t002:** Antibodies.

Target	Dilution	Company	Cat No.
anti-p21^WAF1/Cip1^	1:1000	Cell Signaling, Danvers, MA, USA	#2946
anti-H2AJ	1:1000	Active Motif, Carlsbad, CA, USA	AB_2793769
anti-AR	1:1000	Merck Millipore, Darmstadt, Germany	#06-680
anti-mouse IgG	1:10,000	Cell Signaling, USA	#7076S
anti-rabbit IgG	1:10,000	Cell Signaling, USA	#7074S
anti-β-Actin	1:10,000	Abcam, Ballynew, Ireland	ab6276
anti-AR (ChIP-qPCR)	1:50	Cell Signaling, USA	#5153
E cadherin	1:1000	Cell Signaling, USA	#3195
Vimentin	1:1000	Cell Signaling, USA	#5741s

**Table 3 cancers-17-00791-t003:** AR binding sites to *H2AJ* gene locus (according to Figure 1G)—JASPAR-2024.

Matrix ID	Name	Score	Sequence ID	Predicted Motif Sequence
MA0007.2	MA0007.2.AR	8.886336	Peak region 1	GAGAACATCCACTTA
MA0007.2	MA0007.2.AR	8.77093	Peak region 1	AACAACAGCCTGTCC
MA0007.2	MA0007.2.AR	8.022745	Peak region 1	AGGAACAAGCTCGCC
MA0007.2	MA0007.2.AR	6.7239027	Peak region 2	AGGAACAAATACATA
MA0007.2	MA0007.2.AR	6.207951	Peak region 2	CAGAACATCTCGAAC
MA0007.2	MA0007.2.AR	9.315744	Peak region 3	GGGGACACAAAGACA

**Table 4 cancers-17-00791-t004:** Pathway analysis of *H2AJ* KD transcriptome.

	Fold Enrichment	*p* Value
Signaling by VEGF	1.308	8.72 × 10^−6^
Cellular response to chemical stress	1.951	6.03 × 10^−6^
VEGFA-VEGFR2 Pathway	1.103	5.73 × 10^−11^
G1/S Transition	1.651	2.16 × 10^−7^
DNA Replication	1.497	2.05 × 10^−7^
Signaling by TGFB family members	1.282	0.00011
Signaling by EGFR in Cancer	1.382	0.000242
Circadian Clock	3.421	0.002307
Cellular Senescence	1.625	0.022015
Formation of Senescence-Associated Heterochromatin Foci (SAHF)	1.413	0.028165
Estrogen-dependent gene expression	1.141	0.046655

## Data Availability

The data that support the findings of this study are available from the corresponding author by request.

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
