# Peer review of "H2AJ Is a Direct Androgen Receptor Target Gene That Regulates Androgen-Induced Cellular Senescence and Inhibits Mesenchymal Markers in Prostate Cancer Cells"

_cancers, 2025, doi:10.3390/cancers17050791_

Round 1

Reviewer 1 Report (Previous Reviewer 1)

Comments and Suggestions for Authors

The manuscript titled " H2AJ is a Direct Androgen Receptor Target Gene that Regulates Androgen-Induced Cellular Senescence and Inhibits Mesenchymal Markers in Prostate Cancer Cells" demonstrates involvement of H2AJ which promotes cell growth and senescence while inhibiting mesenchymal marker expression, aligning with its low levels in metastatic tumors. The manuscript is interesting however, it requires following improvements :

  1. How H2AJ levels looks like upon treating cells with AR inhibitors such as Enzalutamide? is H2AJ function as cellular senescence regulator still the same?
  2. Are the CHIPseq peaks represented in Fig. 1g at exons? If yes then Is there any co-relation of these peaks at exons?
  3. Please check if description is missing in table 4.
  4. Also it will be more scientifically gripping if this table is represented as graph.
  5. Please check representation of figure 3a.
  6. Please quantify senescence-associated heterochromatin foci formation in figure 3a.
  7. Authors need to check expression of CDKN1A, VIM, CDH1 and CDH2 upon AR inhibition conditions to confirm regulation of H2AJ via AR.
  8. A deeper discussion of study limitations, challenges, and future research directions, along with a more detailed introduction and discussion, would enhance the manuscript's completeness. To address this deficiency and align with the relevant findings presented in the manuscript, the authors are encouraged to integrate pertinent literature (please refer to PMID: 22403609, 38965223, 32650419). This literature notably delves into the modifications of critical AR associated signaling pathways, elucidating alterations in growth in prostate cancer. By integrating these studies into the manuscript, the authors can substantially strengthen the basis for their compelling and comprehensive concept of H2AJ/AR axis in regulation of senescence in prostate cancer.

Author Response

Reviewer 1:

The reviewer 1 raised now additional points that had not been raised before, which is unusual. Some points being now addressed is to use the AR antagonist Enzalutamide, although this manuscript is focused on the AR agonist signaling by supraphysiological androgen levels used in bipolar androgen therapy.

Nevertheless, we added additional data as follows.

  1. How H2AJ levels looks like upon treating cells with AR inhibitors such as Enzalutamide? is H2AJ function as cellular senescence regulator still the same?

Response: This manuscript specifically focuses on AR signaling regulated by AR agonist at a supraphysiological level (SAL). However, we have now analyzed H2AJ expression by treatment with the second-generation antagonist Enzalutamide. No significant change of H2AJ expression was observed indicating that H2AJ upregulation is rather specific for SAL-induced cellular senescence. We included the novel data. Please see page 7 and supplementary data S1B.

  1. Are the CHIPseq peaks represented in Fig. 1g at exons? If yes then Is there any co-relation of these peaks at exons?

Response: Peaks represented from ChIP-seq data in figure 1G are located upstream and also in exons of the H2AJ gene locus confirmed via ChIP-qPCR experiments. Each of these peaks do contain AREs as identified by JASPAR. The sequences are listed in Table 3.

  1. Please check if description is missing in table 4.

Response: For better clarification the title of table 4 has now been modified. Please see page 17.

  1. Also it will be more scientifically gripping if this table is represented as graph.

Response: Thank you for your comment. The information in this table is derived from the pathway analysis output presented in Figure 5B and includes additional pathways identified in the analysis. Pathways that were not depicted in Figure 5B, but are still of interest, have been included in Table 4.

  1. Please check representation of figure 3a.

Response: We added now higher resolution new pictures. Please see page 13.

  1. Please quantify senescence-associated heterochromatin foci formation in figure 3a.

Response: As shown in figure 3a, we observed changes in SAHF formation following H2AJ KD. However, quantifying the SAHF foci is challenging due to the difficulty in accurately distinguishing and defining the exact boundaries of SAHFs within the nuclei, as foci vary in size and shape in nucleus. Please note, that the overall nuclear staining and fluorescence is not changing, rather foci are generated by higher chromatin re-organization and compaction. Despite this limitation, we believe that the observed changes of SAHF provide valuable insights into the role of H2AJ in cellular senescence and SAHF regulation.

  1. Authors need to check expression of CDKN1A, VIM, CDH1 and CDH2 upon AR inhibition conditions to confirm regulation of H2AJ via AR.

Response: This manuscript focuses specifically on AR signaling regulating cellular senescence in response to AR agonist treatment at SAL. However, we have now analyzed H2AJ expression following Enzalutamide treatment of C4-2 cells (figure S1B). However, no changes in H2AJ levels were observed. These findings suggest that H2AJ is regulated by SAL treatment rather than by AR antagonist. Therefore, further analysis of CDKN1A, VIM, CDH1, and CDH2 under AR antagonist condition is not meaningful, as Enzalutamide treatment does not significantly affect H2AJ expression level.

  1. A deeper discussion of study limitations, challenges, and future research directions, along with a more detailed introduction and discussion, would enhance the manuscript's completeness. To address this deficiency and align with the relevant findings presented in the manuscript, the authors are encouraged to integrate pertinent literature (please refer to PMID: 22403609, 38965223, 32650419). This literature notably delves into the modifications of critical AR associated signaling pathways, elucidating alterations in growth in prostate cancer. By integrating these studies into the manuscript, the authors can substantially strengthen the basis for their compelling and comprehensive concept of H2AJ/AR axis in regulation of senescence in prostate cancer.

Response: Thank you for this valuable comment. We have now provided more information in the discussion regarding the role of antagonists in mediating cellular senescence, as well as other AR-associated signaling pathways involved in this process to strengthen our manuscript and clarify the challenges in AR signaling and its role in cellular senescence in PCa. Furthermore, we have added information to the introduction part to better clarify the gaps in the regulation of cellular senescence in PCa. Please see pages 2, 19, 20, and 21.

Reviewer 2 Report (Previous Reviewer 2)

Comments and Suggestions for Authors

The authors have fully addressed my concerns in the revised manuscript.

Author Response

Thank you to confirm that we have fully addressed your concerns in the revised manuscript.

This manuscript is a resubmission of an earlier submission. The following is a list of the peer review reports and author responses from that submission.

Round 1

Reviewer 1 Report

Comments and Suggestions for Authors

The manuscript titled ‘The Histone Variant H2AJ as a Direct Androgen Receptor Tar-get Gene Regulates Androgen-Induced Cellular Senescence and Inhibits Expression of Mesenchymal Markers in Prostate Cancer Cells" demonstrates that H2AJ, a direct androgen receptor (AR) target, regulates cellular senescence and tumor growth in prostate cancer, while inhibiting mesenchymal transition in prostate cancer.

The manuscript is interesting however, it requires few minor improvements to improve clarity and impact, making it suitable for publication.

1. Please revisit the manuscript title and simplify them for clarity.

2. Please provide the concentration and time duration for SAL treatment to the cells in culture.

3. The authors need to provide scales bares for all the photographic/IHC figures.

4. Please check if figure legend for figure 2 E and D are correct. They looked switched.

5. Please define statistical analysis done for growth assays utilizing cristal violet assay. This information is missing right now.

6. How was CHIP assay performed to generate peaks in figure 1g? authors need to provid that information in method section as well.

7. What are the controls, Authors included in CIP assay in Figure 1G? This information is currently missing in representative peaks.

8. Did Authors validate CHIP peaks as shown in Figure 1G by specific primers through qPCR?

9. how Gene expression correlation between H2AJ and CDH2 drawn in Figure 3F &G? What are statistical tests used?

10. A major limitation of the study is the absence of experiments validating the ChIP-seq data on hormone-dependent recruitment of AR to the H2AJ gene locus in C4-2 cells. Additional experiments are needed to strengthen the robustness of the hypothesis.

Reviewer 2 Report

Comments and Suggestions for Authors

In this manuscript, Horestani and Baniahmad discovered a new AR-target gene, H2AJ, according to RNAseq and ChIPseq data. H2AJ knockdown inhibits prostate cancer (PC) whereas overexpression of it promotes cell growth. This finding should be of some interest for prostate cancer researchers studying AR-regulated genes. However, the manuscript still needs a lot of improvements before can be considered for publication.

1) Fig 1D: The authors should run a SDS-PAGE/WB using protein lysates from all PC cell lines mentioned in this figure, to compare the expression levels of H2AJ (and AR) across different PC cell lines.

2) Fig 1E: SDS-PAGE should be run on the same blot for a proper comparison of DMSO and SAL treatments. Also, protein expression of other known AR target genes (e.g., PSA, TMPRSS2, FKBP5, etc.) should also be shown here. The LNCaP blots are unnecessary here if the authors can provide the WBs that I mentioned in point 1.

3) Fig 1G: There are several AR-binding sites within the H2AJ gene and perhaps its promoter region, but there is no experimental evidence to show which of these binding sites is important for AR to regulate H2AJ expression.

4) Fig 2 is an important figure and therefore this experiment should be similarly carried out on another 2 PC cell lines, e.g., PC cell lines that have high H2AJ expression.

5) Fig 3’s qPCR data only showed changes in mRNA levels, protein level changes of CDH1 and VIM should also be checked by SDS-PAGE/WB.

6) Fig 5’s bioinformatic analyses are poor, there are a lot of missing information here. I would suggest that the authors do a proper GSEA or an over-representation analysis (ORA) for the differentially regulated genes/proteins and clearly state out the cutoffs they used for presenting the data (e.g., NES, FDR, etc.).